# Low Occurrence of Musculoskeletal Symptoms in Swimming? Musculoskeletal Symptoms and Sports Participation in Adolescents: Cross Sectional Study (ABCD—Growth Study)

**DOI:** 10.3390/ijerph19063694

**Published:** 2022-03-20

**Authors:** Santiago Maillane-Vanegas, Francis Fatoye, Rafael Luiz-de-Marco, Jamile Sanches Codogno, Diego Augusto Santos Silva, Carlos Marcelo Pastre, Romulo A. Fernandes

**Affiliations:** 1Laboratory of InVestigation in Exercise (LIVE), Department of Physical Education, Sao Paulo State University (UNESP), Presidente Prudente 19060-900, Brazil; rafaelluizdemarco@gmail.com (R.L.-d.-M.); jamile.codogno@unesp.br (J.S.C.); romulo.a.fernandes@unesp.br (R.A.F.); 2Department of Health Professions, Manchester Metropolitan University, Manchester M15 6GX, UK; f.fatoye@mmu.ac.uk; 3Research Center in Kinanthropometry and Human Performance, Sports Center, Federal University of Santa Catarina, Florianópolis 88040-400, Brazil; diegoaugustoss@yahoo.com.br; 4Department of Physical Therapy, Sao Paulo State University (UNESP), Presidente Prudente 19060-900, Brazil; pastre@fct.unesp.br

**Keywords:** adolescents, sports, musculoskeletal symptoms, swimming, physical activity

## Abstract

The purpose of this paper was to identify the association between the occurrence of musculoskeletal symptoms (MS) and sports participation in adolescents. The sample included 193 adolescents (11 to 17 years of age; 131 boys and 62 girls). For this cross-sectional study, participants were categorized into four groups: “no-sports”, “repetitive non-impact sports”, “high-impact sports”, and “odd-impact sports”. A questionnaire was used, which defined MS as pain or any musculoskeletal complaint that led to restriction of current normal activities. In the entire sample, 112 adolescents reported at least one episode of MS during the recording, representing 58% of the sample. Our findings highlight that adolescents regularly engaged in odd-impact sports, such as martial arts, report a higher occurrence of MS than swimmers and adolescents who do not participate in any physical activity.

## 1. Introduction

In sports, musculoskeletal symptoms (MS) are often identified as injuries, which is not necessarily correct [1]. MS is a complex state defined as “any pain, complaint, or unpleasant sensory and emotional experience associated with or similar to actual or potential tissue damage that leads to restriction of current normal day activities” [1]. In adolescents, MS is a common event (4–60%), which has a significant impact on their future musculoskeletal health [2].

Sports participation is the main manifestation of physical exercise in adolescence; however, together with all the benefits to health variables, sports participation is also commonly associated with the occurrence of MS [3]. Hulsegge et al. (2011), in a large pre-adolescent cohort in the Netherlands, identified that being physically active on at least 5 days a week was associated with a significantly higher risk of MS in lower body extremities [4]. The same phenomenon (occurrence of MS attributed to sports participation) is observed among elite players [5], and girls engaged in sports present a higher intensity of MS than boys [6].

In fact, aspects of sports participation (e.g., training load, sex, amount of training, and also insufficient physical activity) might increase the occurrence of MS. However, some sports seem to increase the risk of MS more than others [6], highlighting the relevance of splitting them according to common aspects, such as biomechanics, and environment where the sport is performed (land or water), among others [6,7]. This aspect was described by Malmborg et al. (2018) in a Swedish cross-sectional study with 178 adolescents, in which contact sports have a higher incidence of injury than non-contact sports and aesthetic sports [7], with the same pattern expected for MS [6].

In fact, physical load is an inherent element in many sports, and is relevant to the risk of adverse outcomes, such as injuries [8]. Therefore, considering aspects of physical load, sports can be divided into levels: high-impact sports [e.g., wrestling and basketball], odd-impact sports [e.g., judo and karate], and non-impact sports [e.g., swimming and cycling] [9]. Moreover, although it is commonly accepted that water sports have less adverse outcomes due to the absence of physical impact, this assumption is not necessarily true for bone fractures [10], and data about MS are still unclear.

The aforementioned context highlights the importance of also recognizing the subjective dimension of the condition, with emphasis on the need to assess MS information and sports participation in adolescents [11]. In light of this, the purpose of this paper was to identify the association between the occurrence of MS and sports participation in adolescents. We hypothesized that MS would be more frequently observed in adolescents engaged in sports compared to those not engaged in sports.

## 2. Materials and Methods

### 2.1. Procedure and Sample

This cross-sectional study is part of the ongoing research entitled “Analysis of Behaviors of Children During Growth—ABCD Growth Study” [12], carried out in the city of Presidente Prudente (~200,000 inhabitants; western region of the state of Sao Paulo, Brazil; Human Development Index of 0.807). The city of Presidente Prudente is in the western region of the state of São Paulo and is characterized as the largest municipality in the region. This characteristic, added to the fact that the city has four colleges and a university within its urban perimeter, allows the city to be a teaching center and increases the chance that the evaluated adolescents will maintain their residence in the city after the end of high school.

Before any fieldwork, the research project was presented to members of the municipal administration (Department of Sports and Department of Education). After approval from the local authorities, researchers contacted coaches (sports clubs) and principals of schools and sports clubs located in the metropolitan area of the city, in order to describe the aims of the research and inclusion criteria, and 11 institutions responded to the invitation. The adolescents were recruited from these included school units (public and private) and sports clubs in the Presidente Prudente region, for which there is already authorization from the Secretariats of Education and Sport Municipalities.

In brief, parents and legal guardians were contacted and only adolescents with a written consent form signed by parents or legal guardians were included in the study (Appendix A). The project was also explained to the adolescents, who each had the freedom to participate in the research or not. The adolescents were then scheduled to visit the university facilities for collection of measurements (DXA, anthropometric data, and face-to-face interview), a process that lasted approximately 30 min per adolescent. The research project was carried out in the first half of 2017.

The sampling process has been described in more detail elsewhere [13,14]. The adolescents were recruited in 11 school units (public and private) and sports clubs (swimming, gymnastics, basketball, baseball, tennis, athletics, karate, judo, and kung-fu) in the metropolitan area of Presidente Prudente.

The sample consisted of 193 adolescents (11 to 17 years old), 131 boys and 62 girls. In total, 127 adolescents were engaged in sports (with a mean age of 14.1; body fatness percentage of 21.4%), and 66 were not engaged in sports (with a mean age of 15.7; body fatness percentage of 23.4%). The amount of time engaged in sports was approximately 598 min per week and the squads were participants in competitions at state and national levels.

For adolescents who were engaged in sports, the following inclusion criteria were applied: (i) previous involvement of at least 12 months in the current sport; (ii) absence of clinical or metabolic disorders (previously diagnosed); (iii) no regular use of medications that may affect the development and growth of the adolescent; (iv) aged from 11 to 17 years; (v) signed written consent form. For adolescents not engaged in sports, the following inclusion criteria were applied: (i) no regular participation in sports in the previous six months; (ii) signed written consent form; (iii) absence of clinical or metabolic disorders (previously diagnosed); (iv) no regular use of medications; (v) aged from 11 to 17 years.

The Ethical Board of the Sao Paulo State University (UNESP) approved the research project (Process number: 02891112.6.0000.5402).

### 2.2. Dependent Variables: Musculoskeletal Symptoms 

#### Musculoskeletal Symptoms

The questionnaire used to assess musculoskeletal symptoms was an adaptation of the frequently used Nordic Questionnaire (Kuorinka et al., 1987, 1990) [15], previously validated for Brazilian Portuguese. The validation study presented satisfactory Kappa index values, ranging from 0.63 to 1.00 [16].

It is worth mentioning that this type of screening tool is often used in the development and assessment of prevention strategies for work-related pain and symptoms [17]. Regarding MS, the questionnaire has previously been used to satisfactorily identify a high prevalence of symptoms in most investigated body parts in a young Brazilian athlete population, especially the knees (50.6%), shoulders (47.5%), and lower back (45.7%). Although this questionnaire is similar to those used in work safety and ergonomics, it could also be useful for school physical activity programs or for individual and team sports [17,18].

The questionnaire presents a scheme of the body subdivided into regions (neck, shoulder, upper back, elbows, wrists/hands, lower back, hip/thigh, knees, and ankles/feet). For each body region, there are three closed dichotomous questions: (i) the presence of any MS in the previous 7 days, (ii) impairment in daily activities over the previous 7 days due to MS, and (iii) consulting a health professional because of MS. For our purposes, we divided the questionnaire a priori between symptoms (yes) of the upper limbs (shoulder, elbow, and wrists/hands), lower limbs (hip/thigh, knee, ankles/feet), and overall musculoskeletal symptoms. The questionnaire was applied by trained researchers and previously explained to avoid any interpretation error on the part of the adolescent.

In the present study, in order to avoid recall bias, researchers considered only the presence of MS in the week before the interview [19]. Additionally, the body segments were clustered into two groups: lower limbs (hip/thigh, knees, and ankles/feet) and upper limbs (neck, shoulder, upper back, elbows, wrists/hands, and lower back), generating the variables MS_lower limbs_ and MS_upper limbs_, respectively. In terms of the outcomes, MS upper limbs, MS lower limbs, and MS overall were treated as categorical (presence of any MS [yes or no]).

### 2.3. Covariates: Sports Participation & Anthropometric Measures

#### Sports Participation

In the face-to-face interview, participants also reported the number of days per week and amount of time per day dedicated to the sport (the data confirmed by the coach), while the no-sports group reported the number of physical education (PE) classes per week (twice a week; 50 min per class). From this, researchers assessed the time of exposure in sports, physical education classes, and recreational physical activities.

Time of exposure in sports and PE classes (in hours) was calculated as follows: (([days per week of organized sport × time per day in hours] × 52 weeks) + ([days per week of PE × time per day in hours] × 52 weeks)), respectively.

The presence of MS_upper limbs_, MS_lower limbs,_ and MS_overall_ per 1000 h of sports-PE exposure was calculated as follows (6): (i) number of MS_upper limbs_/(time of exposure in sports-PE/1000), (ii) number of MS_lower limbs_/(time of exposure in sports-PE/1000), and (iii) number of MS_overall_/(time of exposure in sports-PE/1000), respectively. Thus, the occurrence was expressed as: MS_upper limbs_/1000 h, MS_lower limbs_/1000 h, and MS_overall_/1000 h, respectively.

### 2.4. Anthropometric Measures

Body mass was measured using an electronic scale (Filizzola PL 150, model Filizzola Ltd.a, Brazil with a precision of 0.1 kg). Stature was measured using a stadiometer (Sanny, model American Medical of the Brazil Ltd.a, Brazil, accurate to 0.1 cm). Finally, technical errors of measurement were 0.04% and 0.11% for body mass and height, respectively. All measures were assessed by a single trained researcher.

Biological maturation was estimated by the peak of height velocity (PHV) using mathematical models based on anthropometric measures to calculate the time (in years) remaining (negative values) or past (positive) to PHV, which is an important biological event that follows the human maturation process: Years from age of PHV for males = −7.999994 + [0.0036124 x (Age x Stature)] and Years from age of PHV for females = −7.709133 + [0.0036124 x (Age x Stature)]. Subsequently, PHV was subtracted from the chronological age, giving the age of peak height velocity (APHV) [20]. Participants also reported their engagement in RT.

Whole-body lean soft tissue—LST (in percentage [%] and in kg) and body fatness (%) were assessed using a dual-energy x-ray absorptiometry (DXA) scanner (Lunar DPX-NT; General Electric Healthcare, Little Chalfont, Buckinghamshire, UK) with GE Medical System Lunar software (version 4.7). DXA measurements were performed in the morning after a light breakfast, and the scanner quality was tested by a trained researcher before each day of measurement, following the manufacturer’s recommendations.

Anthropometric measures (weight and stature), engagement in resistance training (RT) (days/week), and body composition (LST and body fatness) were used as covariates.

### 2.5. Statistical Analysis

Measures of central tendency, dispersion, and frequency were used as descriptive statistics. The Student’s t test (continuous data) and chi-square test (categorical data) were used to compare means and to assess associations, respectively. The non-parametric approach (Mann–Whitney) was adopted when the normality assumption was violated. The chi-square test was used to assess the association between sports participation and the presence of MS. The magnitude of the associations was expressed as odds ratio (OR) and its 95% confidence intervals (95% CI) using binary logistic regression. Logistic regression models were adjusted by covariates, while the Hosmer and Lemeshow test assessed the fitness of the multivariate models. Analyses were carried out on statistical software BioEstat (version 5.0) and significance level was set at <0.05.

## 3. Results

Of the 193 adolescents interviewed, the occurrence of MS was not associated with an engagement in sports (Table 1).

### 3.1. Musculoskeletal Symptoms and Independent Variables

A total of 79 MS for upper limbs, 76 for lower limbs, and 112 for MS overall were registered, totaling 267 MS in the last 7 days. The occurrence of MS was not associated with any of the independent variables considered (sex, age, maturation, obesity, and engagement in resistance training) (Table 2).

### 3.2. Musculoskeletal Symptoms and Sports Participation

The repetitive non-impact group were 75% less likely to report any MS in the upper limbs (OR = 0.25 [95% CI: 0.07 to 0.84]) than participants in the no-sports group. Similarly, the repetitive non-impact group were 68% less likely to report MS (OR = 0.32 [95% CI: 0.10 to 0.95]) than participants in the no-sports group. On the other hand, participants in the odd-impact group presented an increased likelihood of reporting MS in the lower limbs than those in the no-sports group (OR = 3.25 [95% CI: 1.25 to 8.47]) (Table 3).

Considering the time of exposure to sports participation, it was observed that participants in the no-sports group reported more MS_upper limbs_/1000 h (9.877 [95% CI: 6.409 to 13.346]) than the repetitive non-impact group (0.393 [95% CI: 0.004 to 0.781]), high-impact group (2.139 [95% CI: 1.346 to 2.933]), and odd-impact group (3.015 [95% CI: 1.316 to 4.713]). Moreover, participants in the no-sports group also reported more MS_lower limbs_/1000 h (5.732 [95% CI: 3.032 to 8.432]) than the repetitive non-impact group (0.382 [95% CI: 0.040 to 0.724]), high-impact group (2.330 [95% CI: 1.409 to 3.250]), and odd-impact group (3.275 [95% CI: 1.818 to 4.731]), and this same pattern was observed in the MS_overall_/1000 h.

The no-sports group reported a higher MS (15.988 [95% CI: 10.902 to 21.075) than the repetitive non-impact group (0.919 [95% CI: 0.283 to 1.554), high-impact group (4.633 [95% CI: 3.108 to 6.159), and odd-impact group (6.542 [95% CI: 3.550 to 9.534]). In addition, the MS_upper limbs_, MS_lower limbs,_ and MS_overall_ report were higher among participants in the high-impact and odd-impact groups than in the repetitive non-impact group. These results are not shown in the tables.

## 4. Discussion

The purpose of this paper was to identify the association between the occurrence of MS and sports participation in adolescents. Our findings identified that adolescents engaged in odd-impact sports (judo, kung-fu, and karate) reported a higher occurrence of MS in the lower limbs, while swimmers reported less MS than adolescents not engaged in sports.

In the entire sample, 112 adolescents reported at least one episode of MS, representing 58% of the sample. This result demonstrates how this phenomenon can be present in both active and non-active populations. Our prevalence is similar to previous observations; for example, El-Metwally et al. (2004) reported prevalences of 45% and 52% for MS in their adolescent cohort over a 3-month recall period [23]. King et al. (2011) found that prevalence rates ranged substantially between the included studies, with MS ranging from 14–24% and multi-site MS ranging from 4–49% [2]. Although the definitions of MS and the logistic processes of data collection may differ, MS is a widely diagnosed condition in adolescents. 

Among adolescents, MS is most commonly reported in the lower extremities [3,24]. In the current study, as in a previous study [25], it was observed that adolescents who participated in odd-impact sports registered more MS in the lower limbs than adolescents that did not engage in contact sports. The highest occurrence of MS in judo (80.1%), followed by kung fu (71.4%), was not entirely a surprise [26], as odd-impact sports appear to be associated with a high risk of traumatic injuries. The impact load is inherent in these full contact sports, playing an important role in the competitive dynamics of the sport and including biomechanical patterns in their practice that increase the risk of these events (e.g., jumping, subtle changes of direction, and receiving impact, among others) [26].

The harmful effect of odd-impact sports on MS in the lower limbs seems to be more attributed to biomechanical aspects than to time of exposure. A negative relationship was observed between time of exposure to odd-impact sports and the registration of MS. Similar findings were observed in the study by Lynch et al. (2017), in which ≥3 days/week and ≥60 min/day of sports participation was related to a lower occurrence of traumatic fractures per 1000 h of exposure to sports participation [27]. Of course, adolescent athletes accumulate more time exposure to sports participation (lower values) than sedentary adolescents [25]. However, it is necessary to consider that although no-sport adolescents presented a lower prevalence of MS, and despite the shorter time of exposure to physical activities and attempts to avoid the appearance of MS due to sports practice, the differences were not distinct between both groups.

Moreover, injuries from overuse appear to be more common in non-contact sports than in odd-impact sports, although in our data the lowest occurrence of MS in swimmers is an interesting finding [28]. This sport modality combines large training loads, high technical requirements, and demands on strength to overcome an external load, mainly in the upper limbs, leading to joint and muscular overloads, increasing the risk of injury [29]. Regarding our data, the apparent protection against MS among swimmers could be attributed to intrinsic factors such as the muscle mass of adolescents, evidencing a protective factor; however, further studies are needed to corroborate this possible hypothesis.

Finally, MS, injury, and illness surveillance is a prerequisite for effective protection of the health of adolescents, providing relevant data on the registration of adverse effects occurring during sports participation [30]. This epidemiological information contributes to the planning and provision of adolescent healthcare, providing objective information and aiding in the development of measures to prevent adverse events, such as safety precautions in sports (e.g., regulations, periodization programs, equipment).

### 4.1. Limitation and Strengths

This study has limitations; first, due to our objectives, the inclusion criteria were highly selective, which could lead to specifically selected participants, meaning it is important to exercise discretion in the generalization of the results and findings. Second, the absence of a medical assessment is a relevant limitation, since the adolescents might underreport less severe MS. Finally, MS can vary between modalities due to the needs or objectives of each sport. For this reason, we segmented the sample to be able to reach the largest number of MS records possible, so it is important to be cautious in the interpretation and generalization of the results.

### 4.2. Practical Implication

For practical implications, MS is present in both adolescents who report sports participation and those who do not; however, adolescents engaged in sports will obtain all the benefits from the practice, enhancing the relevance of motivating the younger population to participate in sports during this period of life. Moreover, swimming seems an attractive option for parents looking for sports participation with a low risk of MS.

## 5. Conclusions

In summary, adolescents regularly engaged in odd-impact sports, such as judo, kung-fu, and karate, reported a higher occurrence of MS in the lower limbs than those engaged in repetitive non-impact sports, such as swimming, and sedentary adolescents. In addition, swimmers reported less MS than other sports and sedentary groups. There was no relationship between time spent on sports participation and number of MS, denoting no risk of MS in adolescents engaged in sports.

## Figures and Tables

**Table 1 ijerph-19-03694-t001:** General characteristics of the sample according to engagement in sports (ABCD—Growth Study; *n* = 193).

	No-Sports (*n* = 66)	Sports Participation (*n* = 127)	
Variables	Mean (SD)	Mean (SD)	*p*-Value
Continuous			
Age (years)	15.7 (2.1)	14.1 (1.9)	0.001
Body mass (kg)	58.0 (13.1)	58.1 (15.3)	0.989
Stature (cm)	167.4 (11.8)	166.3 (11.2)	0.549
BMI (kg/m^2^)	21.32 (3.3)	21.4 (3.9)	0.989
LST (kg) *	41.4 (10.1)	42.4 (10.5)	0.538
Body Fatness (%) *	23.4 (11.8)	21.5 (10.4)	0.262
LST (%) *	71.9 (11.1)	73.9 (11.3)	0.252
APHV (years)	13.8 (0.9)	13.2 (1.1)	0.001
Sport (min/wk)	---	598.8 (391.4)	---
MS (ocurrence) **			
Upper limbs	0.0 (4.0)	0.0 (6.0)	0.341
Lower limbs	0.0 (4.0)	0.0 (5.0)	0.055
MS _overall_	1.0 (7.0)	1.0 (11.0)	0.825
Categorical	*n* (%)	*n* (%)	
RT (yes)	1(1.5%)	38(29.9%)	0.001
MS (yes)			
Upper limbs	29 (43.9%)	49 (38.6%)	0.572
Lower limbs	20 (30.3%)	56 (44.1%)	0.088
MS _overall_	39 (59.1%)	73 (57.5%)	0.951

MS = musculoskeletal symptoms in the last week; SD = standard deviation; APHV = Age of Peak Height Velocity; RT = resistance training; BMI = body mass index; LST = lean soft tissue; CRP = C-reactive protein; * = body composition assessed by DXA; ** = variables expressed as median and range (maximum-minimum) and groups compared using the Mann–Whitney test; ---= non comparison

**Table 2 ijerph-19-03694-t002:** Association between musculoskeletal symptoms and independent variables among adolescents (ABCD—Growth Study; *n* = 193).

		MS_Upper limbs (yes)_	MS_Lower limbs (yes)_	MS_Overall (yes)_
		*n* (%)	*n* (%)	*n* (%)
Sex				
	Girls (*n* = 62)	25 (40.3%)	28 (45.2%)	36 (58.1%)
	Boys (*n* = 131)	54 (41.2%)	48 (36.6%)	76 (58.1%)
	Chi-square	*p* = 1.000 *	*p* = 0.330 *	*p* = 1.000 *
Age				
	11–14 yrs-old (*n* = 101)	41 (40.6%)	42 (41.6%)	59 (58.4%)
	15–17 yrs-old (*n* = 92)	38 (41.3%)	34 (37.1%)	53 (57.6%)
	Chi-square	*p* = 1.000 *	*p* = 0.610 *	*p* = 1.000 *
Body fatness **				
	Non-obese (*n* = 130)	47 (36.2%)	49 (37.7%)	72 (55.4%)
	Obese (*n* = 63)	32 (50.8%)	27 (42.9%)	40 (63.5%)
	Chi-square	*p* = 0.075 *	*p* = 0.595 *	*p* = 0.360 *
Maturation ***				
	Early (*n* = 39)	17 (43.6%)	18 (46.2%)	27 (69.2%)
	On time (*n* = 95)	39 (41.1%)	33 (34.7%)	53 (55.8%)
	Late (*n* = 58)	23 (39.7%)	25 (43.1%)	32 (55.2%)
	Chi-square	*p* = 0.706 *	*p* = 0.913 *	*p* = 0.207 *
RT				
	No (*n* = 154)	64 (41.6%)	61 (39.6%)	91 (59.1%)
	Yes (*n* = 39)	15 (38.5%)	15 (38.5%)	21 (53.8%)
	Chi-square	*p* = 0.866 *	*p* = 1.000 *	*p* = 0.681 *

MS = musculoskeletal symptoms in the last week; RT = resistance training; * = chi-square test; ** = body fatness ≥ 25% for boys and ≥ 30% for girls [21]; *** = Werneck et al. Childhood Obesity 2016 [22].

**Table 3 ijerph-19-03694-t003:** Association between musculoskeletal symptoms and sports participation among adolescents (ABCD—Growth Study; *n* = 193).

	Chi-Square Test (χ^2^)	Binary Logistic Regression *	
Independent Variable	*n* (% [95% CI])	OR_crude_ (95% CI)	OR_adjusted_ (95% CI)
		Outcome: MS_Upper limbs (yes)_	
No-sport	30 (45.4 [33.4 to 57.4])	1.00 (Reference)	1.00 (Reference)
Repetitive non-impact	05 (20.1 [4.3 to 35.6])	**0.30 (0.10 to 0.89)**	**0.25 (0.07 to 0.84)**
High-impact	29 (42.1 [30.3 to 53.6])	0.87 (0.44 to 1.71)	0.65 (0.27 to 1.56)
Odd-impact	15 (45.5 [28.4 to 62.4])	1.00 (0.43 to 2.31)	0.95 (0.37 to 2.43)
*p*-value	0.921 **	-	0.516 ***
		Outcome: MS_Lower limbs (yes)_	
No-sport	20 (30.3 [19.2 to 41.3])	1.00 (Reference)	1.00 (Reference)
Repetitive non-impact	05 (20.1 [4.3 to 35.6])	0.57 (0.18 to 1.74)	0.63 (0.19 to 2.14)
High-impact	32 (46.4 [34.6 to 58.1])	1.98 (0.98 to 4.03)	2.27 (0.91 to 5.65)
Odd-impact	19 (57.6 [40.7 to 74.4])	**3.12 (1.31 to 7.43)**	**3.25 (1.25 to 8.47)**
*p*-value	0.003 **	-	0.556 ***
		Outcome: MS_Overall (yes)_	
No-sport	39 (59.1 [47.2 to 70.9])	1.00 (Reference)	1.00 (Reference)
Repetitive non-impact	08 (32.1 [13.7 to 50.2])	**0.32 (0.12 to 0.86)**	**0.32 (0.10 to 0.95)**
High-impact	43 (62.3 [50.8 to 73.7])	1.14 (0.57 to 2.28)	1.02 (0.42 to 2.48)
Odd-impact	22 (66.7 [50.5 to 82.7])	1.38 (0.57 to 3.32)	1.40 (0.53 to 3.69)
*p*-value	0.301 **	-	0.611 ***

OR = odds ratio; 95% CI = 95% confidence interval; MS = musculoskeletal symptoms in the last week; * = model adjusted by sex, chronological age (baseline), age at peak height velocity (baseline), whole-body lean soft tissue (baseline), and resistance training (baseline); ** = chi-square test; *** = Hosmer and Lemeshow test. Bold: Statistical difference found

## Data Availability

The data used to support the findings of this study are available from the corresponding author upon request.

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
