# Peer review of "Low Occurrence of Musculoskeletal Symptoms in Swimming? Musculoskeletal Symptoms and Sports Participation in Adolescents: Cross Sectional Study (ABCD—Growth Study)"

_ijerph, 2022, doi:10.3390/ijerph19063694_

Round 1
Reviewer 1 Report
The topic of the present study is relevant for sport science and public health. However, this manuscript has serious methodological problems that should be solved.
The aim of the study was to “analyze the comparisons between different impact sports modalities and the presence of musculosceletal pain (MS) in adolescents, as well as to identify the relationship between the occurrence of MS and healthcare costs”. The aim should be revised since it is impossible to understand what does it mean “to analyze comparisons”. The meaning of “different impact sports” was never presented and explained in Introduction. What were hypotheses for this research?
Introduction should be revised and clearly focused to the main problem. It is unclear why authors decided to analyze healthcare costs and how they understand it. Is it personal or public healthcare costs? The rationale for study should be presented in light of the other research. What new this research gives for international science? The classification of sports should be presented in Introduction since not all readers would understand what is repetitive impact, high impact, odd – impact sports. Epidemiology of MS in each category should be presented.
The procedure of the research is not presented. It is unclear who, when and how implemented interviews, how the sample was achieved, especially sports participating adolescents? Could adolescents leave the procedures if they wanted to quit the interview or other parts of research? How many adolescents refused to participate or left research? Who when and how implemented measurements? How long it took to participate in whole research for each adolescent? How somatic maturation was measured? How was engagement in resistance training measured?
The categorization of sport groups should be explained and referencing for it should be provided (repetitive impact, high impact, odd – impact groups)?
How was obesity classified?
Instruments used in this study should be presented clearly. Why adaptation of previously used definitions was necessary in the present study (p.3, lines 113-119)?
Assessing MS pain authors state that they used Nordic Questionnaire previously adapted for Brazilian Portuguese language and cite ref. 9 and 11. Is this correct? Reference no. 9 presents adaptation of the Extended Nordic Musculoskeletal Questionnaire (NMQ-E) for younger populations and adaptation in French language. Reference 11 - analyses the burden of low back pain (LBP) in 17 year olds in terms of specific LBP related impact and general HRQOL. Please provide reference and statistical data reporting validation of Nordic Questionnaire instrument in your national language.
It is also unclear if information in p. 3, lines 128-137 part of Nordic Questionnaire or this was separate questions?
Why was decided to include information about MS the week before the interview?
The disproportion of sports not involved and involved adolescents is too big, is it statistically correct to compare such different groups?
Is mean BMI = 58,0 and 58,1 kg/m2 correct? (Table 1).
What does it mean “upper limbs” and “lower limbs” in Table 1? Was it “MS in upper limbs”?
The counting of amounts of sport participation presented in methods and in Table 1 differs. This information should be adjusted.
Conclusion “there was no relationship between healthcare costs and musculoskeletal symptoms” is unclear and should be revised.
Author Response
Please see the attachment. Regards

Reviewer 2 Report
Thank you for the opportunity to review this manuscript. The purpose of the study was to investigate the presence of musculoskeletal symptoms and associated healthcare costs in adolescents participating in three different impact sports and a control group obtained by a questionnaire.
I can see the interest of the topic for the readership of the scope of IJERPH.
However, I have to address some serious comments that should be addressed before the manuscript can be considered for potential publication.
The quality of your English needs to be revised throughout the manuscript. The draft needs to be revised by a native speaker.
Abstract/Title:
Sorry, but the quality of your Abstract is quite poor. There are serious flaws in writing style and also in consideration of the Abstract’s structure. You have to revise this part thoroughly. Please, stay within the clear structure: Objective, Methods, Results, Discussion and/or Conclusion. Be clear in your writing style, especially, according to sentence length and grammar.
Keywords:
Please, revise your keywords; they do not fully support the main take-aways of your study.
Manuscript:
Introduction:
General comments:
You need to thoroughly revise the whole Introduction, especially in terms of English grammar and writing style.
Specific Comments:
LL32: You are reporting outcomes of “pediatric” populations. I doubt this is the same peer as adolescents. Do you have findings of relevant peers – here adolescents.
L37-39: Please, grammatically revise sentence for better understandability.
L40: “musculoskeletal pain” appears to be an example for musculoskeletal symptoms. Therefore, introduce with: e.g., in the parentheses. Furthermore, you should not give to parentheses next to each other.
L41-44: Example here: Do not try to put too much content in one sentence. It is very hard to generally understand and follow your thoughts. Reduce sentence length in general.
L47: I do not see the link to “pediatrics” when having a age ranging of 5 to 24 years.
L48: See comment above related to the findings of the Canada population.
L52-54: I do not see the full clear link between the sensitivity for pain recognition in pediatrics due to the pain and having poor health in adult life. Be more precise in the report of the results.
L54-56: Okay, you give the numbers here, but I would suggest changing the arguments for a more logic connection.
L61-62: Revise sentence.
L62-65: Sentence appears to be incomplete.
L70 et seq.: Please, clearly elaborate the purpose and the objective of your study.
Methods
I do not see the reason for blind-folding the city and area.
L88-90: Revise English.
L91-93: Remove digits in parentheses!
L93-96: Based on which assumption did you do the grouping of the study participants. If building such groups, the reader needs more information about the recruitment process as well as about the sub-group characteristics, as mass, body height, time of activity in the relevant sport (in years), time participating in sports (hours per week). Table 1 that contains these facts needs to be part of the Methods, for me, not the Results Section.
L103-104: I would say inclusion criteria (iv) and (v) a contradictory. I’d assume that one appears to be the inclusion criteria for the study groups (iv) and (v) is the inclusion criteria for the control group. You should separate the inclusion criteria for the respective groups.
L113-117: Clarify that statement. It appears very convoluted. And: Did the participants know in what kind of study they participated? If so, I’d assume there is high risk of having a participant-related bias in the data. As pain is a highly individual feeling, I’d assume that, especially adolescents, are at risk for giving expected or unexpected answers towards the stated questions – and the individual mood can play an important role in modifying the result. Did you care in any way for that fact?
L117-120: Same here.
L121 et seq.: Please, elaborate this section clearer and give us information about the reliability, objectivity and validity of the questionnaire.
L128-130: This selection criteria is highly at risk to produce errors, in terms of having a directed research of specifically selected participants – this is fair but you should need to care for this fact towards the whole manuscript and especially towards the generalizability of your results and findings. Those are highly selective towards region and the sample you have recruited.
L132: You do not need to explain the term dichotomous in a scientific paper.
L132-135: These are no questions from my point of view – rather the questionnaire’s categories. I would recommend to stay close to the original source of the questionnaire in describing.
General to that section: Is this an already validated and reliable questionnaire? If so, you need to comment that. If not, you should provide some strong arguments, why this questionnaire is actually valid to scientifically answer the related research questions.
L139-140: What is the rationale for that assumption?
L141-143: I am at doubt that the rather arbitrary attribution of the segments to a body region is well chosen. Consider to include a separate trunk region, instead of relating all the mentioned regions to the upper limb region.
L144-147: You need to clarify that part in terms of unproper description.
L149-152: Is there no official information to get about general costs for healthcare in Brazil? I mean to the worldwide ICD system, there should be at least sums for a whole ICD.
I would argue that there is a high risk for bias if you just estimate the costs by knowing which therapy they received. You should give at least, then, an overview what the mentioned interventions or therapeutical actions cost in general.
Additionally, as you did need to convert the currencies, there is another big issue towards the numbers you’ve analyzed. I need more information about the fact, how you did control all the vague assumptions and estimations.
L163-181: Try to reduce section length.
L183: I never met the criteria in anthropometrics “stature”. How did you measure that? Give mass and/or BMI and we do know enough. And: why is trunk height important?
Results
General: Could you please report your results in a standardized manner, please? So, from no sports group to high impact sports group. And keep this structure throughout the section. Then, it is easier to follow you, because you have lots of results to report. Structuring helps a lot.
Table 1: How do you explain the extremely high standard deviations in the variables “upper limbs”, “lower limbs” and “MSoverall”?
Accordingly, you should be consistent with the variable definition in terms of “Upper Limbs” / “Lower Limbs” / and “MSoverall”.
L217-219: I cannot recall these numbers from your Table 1.
Table 2: Try to edit your Tables more homogenous.
Here you also give numbers about somatic maturation. However, you did not mention this outcome variable before nor is it clear why you actually collected those data and how that information did influence the grouping or results.
L248-249: This sentence appears to be incomplete.
Important: Consistency of result reporting. In Table 4 you abbreviate musculoskeletal symptoms as MKS; before-hand as, solely, MS. I don’t get this.
Discussion
General: Try to re-structure your Discussion towards a clearer storyline. It is a bit hard to follow your clear thoughts, findings, and conclusions. Please revise, accordingly.
L268-270: Revise sentence grammatically.
L272: “Adolescents”; in the Introduction often “pediatric”. I mentioned this before. Be more consistent.
L274-276: Is that really a new result that people who do sports have more MS than those who don’t attend in any institutionalized sports? And these results could imply, that doing sports raise a larger economic burden than doing no sports – although we know that doing no sports over the life-span leads to way higher healthcare costs, because of obesity and other metabolic diseases.
L279: Is it really reasonable to compare the results of your sample to the results of a study investigating Ethiopian and Thai population? Why did you choose those? Are there no results of societies similar to yours?
L301-308: I do not see the link of your results to those assumptions about swimmers. What would happen if swimmers would receive impacts? Wouldn’t there to expect also lots MS symptoms. I think it would be of high value to relate the impacts to hours that the athletes participated in sports to receive a comparable metric. Furthermore, to compare to swimmers does not fully proof the fact for me in terms of sports-related MS symptoms. Maybe the reduced gravity during swimming could be a reason for some of the findings. Therefore, I see big problems in the comparison of the different sports. Why did you actually choose this sport as non-contact sport? There would be actually lots others that do not have any opponent-related impacts, e.g., runners or bicycle rides.
L297-300: These assumptions need referencing.
L344: I would say, you’re submitting your manuscript for potential publication in a scientific journal. If so, you should have sufficient reasons in terms of the strength of your paper already described until that point.
L345: Fair argument, but actually, why did you than only chose this one point for your investigation. I would say, you totally need to refer your results and findings to the timepoint of investigation!
Conclusions
Please, completely revise this section. You should give some practical implications of your findings beyond the part of just repeating your results. How do your results enlarge the knowledge towards sports and occurrence of musculoskeletal symptoms?
Thanks for your work!
Author Response
Please see the attachment, regards

Round 2
Reviewer 1 Report
Authors addressed majority of my comments, however, there are some important details that should increase the quality of manuscript:
Sample in terms of sports participation, age (mean age, age range), gender should be presented in abstract.
Please change „physical exercise“ to „physical activity“ in the first sentence of the manuscript (p. 1, line 28).
Please present the term „musculoskeletal symptoms“ first and use abbreviation MS after.
The terms musculoskeletal symptoms, musculoskeletal injuries, musculoskeletal problems, musculoskeletal complaints were used interchangeably across the manuscript. Please present the definition of MS in introduction instead of Methods (see p. 3, lines 103-109) and use one term MS across all text.
Presentation of the sample should include numbers of boys, girls, mean age, age range, differences of these variables in sports participating and not participating groups. Number of overweight and obese adolescents across sport participation groups should also be included. I suggest deleting this information from Table 1 and include it in text (sample presentation in Methods).
It is still not clear how the sample was achieved – please insert 1 paragraph of your response (no 10) to the reviewer in Methods section.
The subsection 2.1. should be renamed as Procedure and sample.
It is unclear what measure is for Body Fat (%) and LST (%) in groups of sport participation (mean (SD) in Table 1? Is it number and percent? What is 23.4 for body fat in Table 1?
Practical implication should not be included in Conclusion section (p. 9, lines 357-362). Please also consider deeper discussion on practical implications of your study. You might discuss specific injury and musculoskeletal symptoms prevention in particular sports not only general PA promotion in adolescents.
Reviewer 2 Report
Dear authors,
Thanks for the work you put into the revision of the manuscript. However, I’ve some lingering comments and questions that need to get addressed before I can accept the manuscript for potential publication.
Affiliations:
Your affiliations are incomplete!!
Abstract:
You need to introduce abbreviations in the Abstract and throughout the manuscript on the first occasion. You are not working pretty clean on that side. MS can mean such a lot of things.
Manuscript:
Introduction:
General comments:
I see an improvement in your Introduction. However, I do not see the full rationale for the study’s purpose? Why did you actually compare those groups? Is there any reason, why you actually compare those “impact groups”?? They appear quite arbitrary taken. How is that comparable to studies of Denmark and Canada, you tell the numbers, but what did they actually investigate? For me, there are still serious flaws in the reasoning for your study purpose. At one point you argue about the “impact-nature” of a sport, on the other hand, you talk about engagement. These are different viewpoints. And if you hypothesize towards engagement, why did you build groups on the “impact nature” of a sport?
“the purpose of this paper was to identify the relationship between the occurrence of musculoskeletal symptoms, healthcare costs and sports participation in adolescents”
This is no clearer elaboration of the study’s purpose.
Tables
You need to fully revise your Results Tables. Tables need to have stand-alone understandability. I do NOT understand those! Be consistent in the structure! Be clear with your captions. Asterisks are quite arbitrary used: At one point it stands for statistical computation, at another for threshold value, and as well for a reference. This totally confuses me. The editing needs improvement as well.
Discussion
After connecting your study to others in Canada and Denmark, here compare to Thai and Ethiopian societies, without having mentioned those studies before. This is only one example, but you need to discuss references from the Introduction here and not just pick other ones that fit better to your Results. Please, put more effort into a neatly-prepared manuscript that follows well-scientific work.
It is so hard to follow your you, as I miss the clear structure of how a scientific paper is set up.
Please, revise your manuscript in terms of that all your arguments and discussion points are always linked to your research questions.
Round 3
Reviewer 1 Report
Authors addressed comments and suggestions.
Reviewer 2 Report
Thanks authors for the additional work you put in the enhancement of your manuscript.
Abstract is fine for me now.
The Introduction is way more fluent to read and to follow. However, there some grammatical errors, as missing verbs at some occasions as well as some type-spelling errors. Please, thoroughly revise the Introduction on that side again.
I still miss the clearly elaborated research gap that your research wants to adress but your argumentation towards your purpose is acceptable.
Tables are alright now, although containing plenty of information and I'm not fully convinced that this all necessary. But it's no need to further complaining.
Discussion is alright, however, I still miss the repetition of your purpose at the beginning, before telling me your major outcome(s).
L300-301: This is not really discussing your outcomes with the outcomes of other studies. Well, you should discuss, why your results may differ to others or why they are similar. Discussing is a method to interpret results and why they appeared as they did, with all pros and cons.
"Thus, MS seems a relevant outcome among adolescents" is just not reasonable statement in a discussion for me.
The second paragraph is better. You should at least discuss your arguments of the first discussion paragraph similarily.
L332-339: I just do not get the logic connection of those arguments. Do you have any arguments, why high-impact sports and non-particpation leads to lowest healthcare costs? This is an odd finding for me and needs thorough interpretation. Further, the connected studies do not fully link to your results, don't they?
Your conclusions and practical implications are vague based on your outcomes.
